# Polyamide membranes with nanoscale ordered structures for fast permeation and highly selective ion-ion separation

Changwei Zhao [1,5] ✉, Yanjun Zhang[1,5], Yuewen Jia[2,5], Bojun Li [3,5], Wenjing Tang[1], Chuning Shang [2], Rui Mo [1], Pei Li [4], Shaomin Liu [4] & Sui Zhang [2] ✉

Fast permeation and effective solute-solute separation provide the opportunities for sustainable water treatment, but they are hindered by ineffective membranes. We present here the construction of a nanofiltration membrane with fast permeation, high rejection, and precise $Cl^-/SO_4^{2-}$ separation by spatial and temporal control of interfacial polymerization via graphitic carbon nitride ($g$-$C_3N_4$). The $g$-$C_3N_4$ nanosheet binds preferentially with piperazine and tiles the water-hexane interface as revealed by molecular dynamics studies, thus lowering the diffusion rate of PIP by one order of magnitude and restricting its diffusion pathways towards the hexane phase. As a result, membranes with nanoscale ordered hollow structure are created. Transport mechanism across the structure is clarified using computational fluid dynamics simulation. Increased surface area, lower thickness, and a hollow ordered structure are identified as the key contributors to the water permeance of 105 L m²·h⁻¹·bar⁻¹ with a $Na_2SO_4$ rejection of 99.4% and a $Cl^-/SO_4^{2-}$ selectivity of 130, which is superior to state-of-the-art NF membranes. Our approach for tuning the membrane microstructure enables the development of ultra-permeability and excellent selectivity for ion-ion separation, water purification, desalination, and organics removal.

Nanofiltration (NF) is a cost-effective and environmentally friendly separation method used in municipal wastewater treatment, drinking water purification, and pharmaceutical purification[1–5]. Compared to other technologies, NF consumes lower energy, takes up less space, and can effectively separate ions. A durable membrane with desirable permeability and selectivity is essential for establishing the ideal NF process. A high membrane permeability allows for operation at lower pressures, further reducing the energy cost. Another important criterion is selectivity. Many studies to date have emphasized the selectivity of water over solutes in NF applications[6]. In recent years, solute-solute separation has attracted increasing attention as a value-added process to water treatment, which may enable many possibilities, such as the energy-effective removal of a specific solute from the water stream and isolation and concentration of a valuable component[7]. One typical example is the separation of chloride from sulfate in the chloralkali process. $Cl^-/SO_4^{2-}$ separation is beneficial for recovering valuable salt resources and obtaining clean freshwater[8–13].

[1]College of Resources and Environmental Sciences, China Agricultural University, Beijing 100193, China. [2]Department of Chemical and Biomolecular Engineering, National University of Singapore, 4 Engineering Drive 4, 117585 Singapore, Singapore. [3]State Key Laboratory of Water Environment Simulation, School of Environment, Beijing Normal University, Beijing 100875, China. [4]College of Materials Science and Engineering, College of Chemical Engineering, Beijing University of Chemical Technology, 100029 Beijing, China. [5]These authors contributed equally: Changwei Zhao, Yanjun Zhang, Yuewen Jia, Bojun Li. ✉e-mail: zhaocw@cau.edu.cn; chezhasu@nus.edu.sg

NF membranes offer a wide range of applications due to their diversified pore size and charge characteristics[14–17]. Their molecular weight cut-off (MWCO) typically ranges between 200 and 1000 Da, and membrane surfaces can be positive, negative, neutral, or zwitterionic. The design of NF membranes with appropriate microstructure, thickness, pore size, and surface charge should be tailored to suit each specific application. Most commercial NF membranes are now manufactured by the interfacial polymerization (IP) of piperazine (PIP) in the aqueous phase and trimesoyl chloride (TMC) in the organic phase[18–20]. Polyamide-based thin film composite membranes remain the gold standard in both the market and academic research[21,22]. The quick and less regulated reaction, however, makes it challenging to tune the structure of the resulting polyamide layer[23,24]. As a result, NF membranes are subjected to an inherent trade-off between flux and selectivity.

Many recent efforts have been made to enhance the permeability by experimenting with various reaction conditions, such as modifying the compositions of the aqueous and organic phases, incorporating nano-fillers, and employing layer-by-layer and 3D printing technologies[25–31]. These strategies seek to manage the IP process by either slowing down the reaction or confining the reaction in a limited space[6]. As a result, the impact on membrane structure and separation performances is limited. Furthermore, most studies do not address solute–solute selectivity.

Herein, a graphitic carbon nitride (g-C$_3$N$_4$) nanosheet was adopted during IP to form an ordered nanoscale hollow structure of a thin-film composite membrane. Molecular dynamics (MD) investigations revealed that g-C$_3$N$_4$ plays a critical role in temporally controlling the IP process by slowing PIP diffusion and spatially restricting the reaction via tiling at the oil–water interface. PIP diffusivity was reduced by one order of magnitude in experiments, and an NF membrane with a nanoscale-ordered hollow cone structure was created, the size and distribution of which were depending on g-C$_3$N$_4$ concentration. The

impact of such a structure on the transport mechanism was investigated further. The membrane exhibited high permeance and outstanding Cl$^-$/SO$_4^{2-}$ selectivity.

Notably, g-C$_3$N$_4$ has been used in prior studies as nanofillers[32] or interlayer[33] during IP. In this work, we prepared g-C$_3$N$_4$ suspensions with high concentrations that induce spatial and temporal modulation to get a nanoscale-ordered morphology and extraordinary separation capacities, and the role of g-C$_3$N$_4$ was systematically studied using molecular simulation. Hence, this research demonstrates an approach to realizing simultaneous temporal and spatial control of reactions and to overcoming the trade-off between permeability and solute-solute selectivity. It provides insights into the design of highly ultra-permeable and selective NF membranes in water purification, desalination, and resource recovery.

## Results and discussion

### Temporal and spatial modulation of reaction by g-C$_3$N$_4$

The key to achieving an ordered structure for ultra permeability and precise separation is to control the IP process, including diffusion and reaction. Here, Fig. 1 illustrates the reaction principle of PIP and g-C$_3$N$_4$ nanosheet (see Supplementary Fig. 3 and 4) with TMC for preparing the membranes. In the IP reaction process (see Supplementary Fig. 5), amines are dissolved in water, and acyl chlorides are dissolved in an organic solvent. The functional layer forms (Fig. 1a) on top of a porous PES support (see Supplementary Fig. 6) or as a freestanding film (see Supplementary Figure 7) after the two immiscible phases' contact.

The diffusion coefficient of PIP is about $10^{-10}$ m$^2$ s$^{-1}$ by nuclear magnetic resonance (NMR) measurement. When g-C$_3$N$_4$ was added into the PIP solution, the monomer diffusivity decreased to $10^{-11}$ m$^2$ s$^{-1}$, one order of magnitude lower than its original value. By real-time online optical monitoring (see Supplementary Fig. 8), it is observed that the spreading area of the (PIP + g-C$_3$N$_4$)/TMC system (Fig. 1b) is smaller than that of the PIP/TMC system within the same time (Fig. 1c),

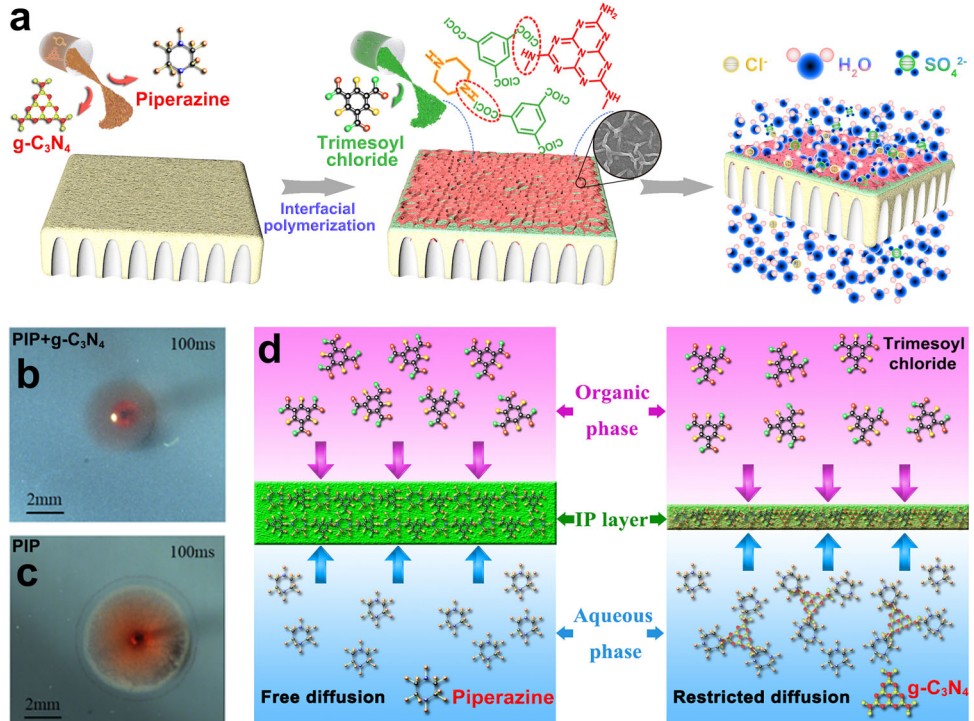

**Fig. 1 | Preparation of NF membranes regulated by g-C$_3$N$_4$. a** Schematic illustration of the g-C$_3$N$_4$-regulated membrane preparation process. **b** Optical photograph capturing the PIP + g-C$_3$N$_4$ + TMC reaction at 100 ms. **c** Optical photograph capturing PIP + TMC reaction at 100 ms. **d** Illustrations comparing the free and restricted diffusion of PIP during the IP process with and without g-C$_3$N$_4$.

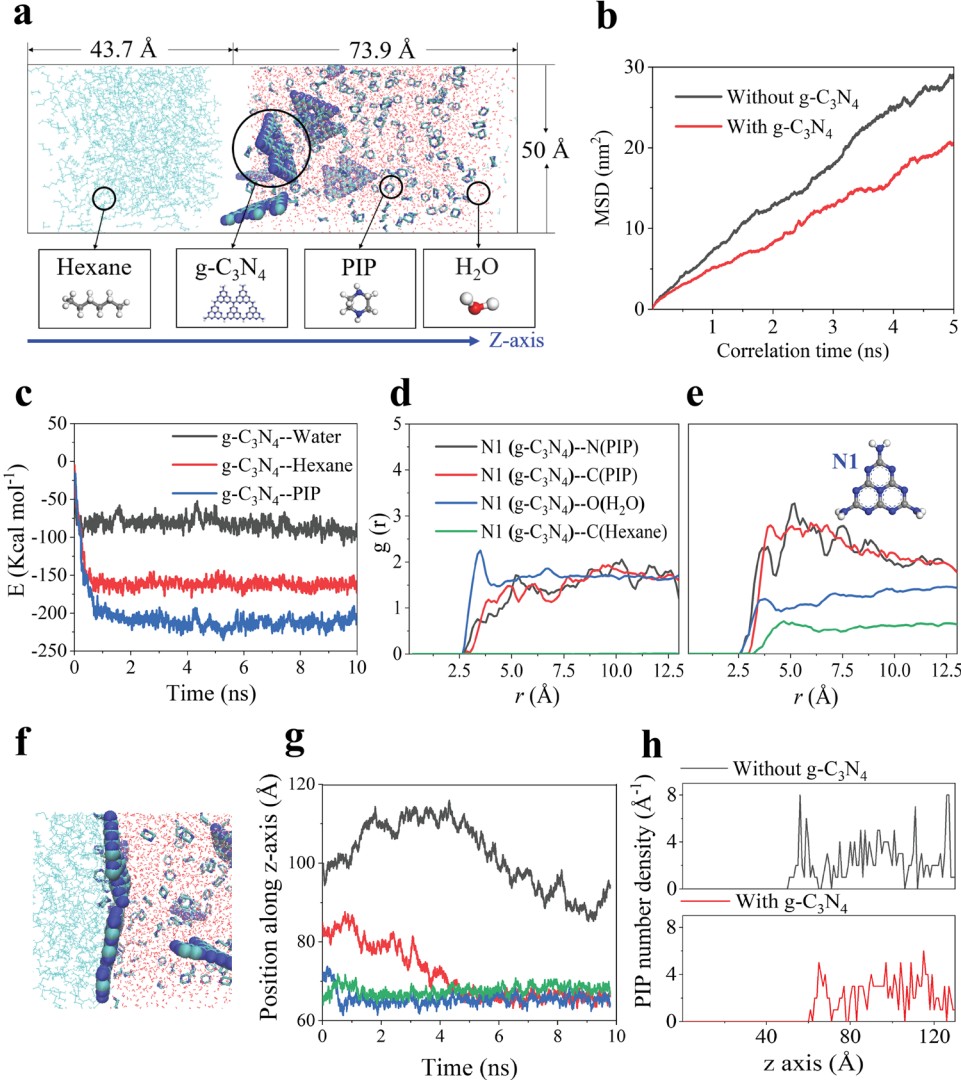

**Fig. 2 | Molecular dynamics simulation of the interfacial polymerization process. a** Simulation models: the left chamber contains hexane and the right chamber is filled with water, PIP molecules, and g-$C_3N_4$; a similar model without g-$C_3N_4$ was also constructed as control. **b** Time dependence of MSD curves of PIP molecules. **c** Interaction energy between g-$C_3N_4$ and other species calculated by model B (see Methods). **d** The RDF of N1 atoms in g-$C_3N_4$ around PIP (N, C), water (O), and n-hexane (C) at the start of the simulation. **e** The RDF of N1 atoms in g-$C_3N_4$ around PIP (N, C), water (O), and n-hexane (C) at the end of simulation; **f** Image capturing the tiling of g-$C_3N_4$ at the water-hexane interface at the end of the simulation. **g** Time dependence of the position of four pieces of g-$C_3N_4$ nanosheets along the z-axis. **h** the PIP number density along the z-axis in the simulation systems. Z-axis starts from the hexane phase to the water phase, as is captured in 2a. Source data are provided as a Source Data file.

which confirms that the mobility of PIP within the (PIP + g-$C_3N_4$)/TMC system is lower than that with the PIP/TMC system. Hence, the diffusion rates of the amino monomers may be adjusted by tuning the g-$C_3N_4$ to reach an ideal difference in the diffusivities of monomers in both the aqueous and organic phases[29] (Fig. 1d).

MD simulations were performed to better understand the roles of g-$C_3N_4$ during the formation of polyamide thin film (see Supplementary Fig. 1). Figure 2a captures model A, which consists of a hexane phase, and an aqueous phase containing water, PIP, and g-$C_3N_4$ nanosheets. A control model without g-$C_3N_4$ was also constructed. The diffusivity of PIP molecules was qualitatively determined by their mean square displacement (MSD) during simulation since the slope of the linear regression between MSD and simulation time is proportional to the dynamic movement of PIP[34]. Figure 2b shows that PIP diffusion significantly decreases in the presence of g-$C_3N_4$. This might be partially explained by the strong affinity between g-$C_3N_4$ and PIP, which is evidenced by the lowest interaction energy between g-$C_3N_4$ and PIP compared to that with water or n-hexane in Fig. 2c, calculated based on

Model B (see Supplementary Fig. 9). It is also noteworthy that g-$C_3N_4$ has a higher tendency to interact with n-hexane rather than water. The radial distribution functions (RDF, $g(r)$) of one representative atom in g-$C_3N_4$, named N1, around different atoms in PIP, water, and n-hexane at the start (Fig. 2d) and end (Fig. 2e) of the simulation provide further insights. Starting from a random status, the peak height for N1–$H_2O$ is reduced, and N1 becomes closely surrounded by the nitrogen atoms in PIP at the end of the simulation, likely through hydrogen bonding. A similar trend is observed with the C atom in g-$C_3N_4$ (see Supplementary Fig.10). It suggests that the hydrogen bonding between g-$C_3N_4$ and PIP may contribute to strong interactions, thus retarding the diffusion of PIP, which is in agreement with experimental observations. This will subsequently lead to better temporal control of the IP process.

In addition, Fig. 2d shows that though originally N1 stays too far away from hexane to be captured by the measurement, they get as close as N1-water at the end. This is evidenced by an interesting tiling behavior of nanosheets at the water-hexane interface, as presented in

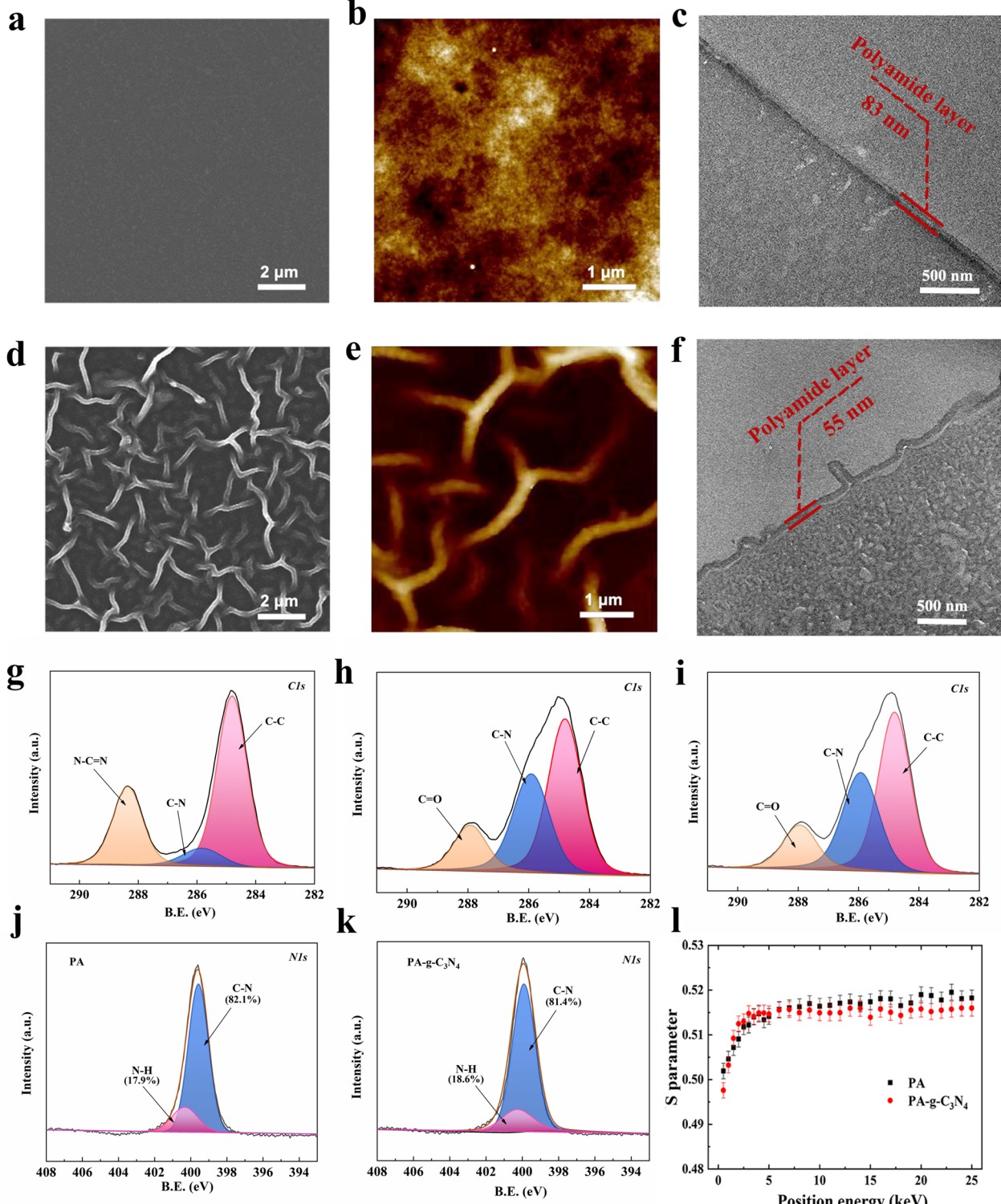

**Fig. 3 | Characterizations of NF membranes. a** The SEM surface morphology of the PA membrane. **b** The AFM image of the PA membrane. **c** The TEM morphology of PA membrane. **d** The SEM surface morphology of the PA-g-C$_3$N$_4$ membrane. **e** The AFM image of the PA-g-C$_3$N$_4$ membrane; **f** The TEM morphology of the PA-g-C$_3$N$_4$ membrane. **g**–**i** C1s XPS spectra for g-C$_3$N$_4$, PA, and PA-g-C$_3$N$_4$ membrane respectively. **j**, **k** N1s XPS spectra for PA and PA-g-C$_3$N$_4$ membrane respectively. **l** S parameter as a function of positron energy for PA and PA-g-C$_3$N$_4$ membrane. The concentration of g-C$_3$N$_4$ for membrane preparation was 0.135 wt%. Source data are provided as a Source Data file.

Fig. 2f. Due to their preferential interaction with hexane over water, the g-C$_3$N$_4$ nanosheets migrate quickly to the interface along the *z*-axis (Fig. 2g) and pave the interface as flooring tiles. Furthermore, the edges of g-C$_3$N$_4$ could be decorated with some amine groups, which is proven by the X-ray photoelectron spectroscopy (XPS) result in

Supplementary Figure 4b and also captured in its structure in MD simulation. The nanosheets may therefore react with TMC to form a relatively solid 'floor'. As a result, PIP can diffuse from the water phase to the interface through the slit between adjacent nanosheets or the inter-plane pores. Both the close PIP-g-C$_3$N$_4$ interactions and interface

tiling slow down the diffusion of PIP, leading to a much lower PIP concentration at the interface (Fig. 2h). Tiling also affects the movement of PIP in space, thus controlling the reaction spatially.

## Polyamide membranes with nanoscale-ordered hollow structure

Figure 3 presents the membrane morphology and properties. The PIP-TMC polyamide membrane surface is smooth (Fig. 3a–c). By adjusting g-$C_3N_4$, nanoscale-ordered hollow structures (Fig. 3d–f, Supplementary Figs. 11 and 12) are obtained. The emergence of ordered structure might be attributed to the retarded diffusion of PIP in the presence of g-$C_3N_4$, similar to the local activation and lateral inhibition phenomenon reported in earlier studies[23,35,36]. However, it means that the number of arched channels increases as g-$C_3N_4$ content rises until reaching 0.135 wt% (Supplementary Figures 12). This is well explained by the tiling effect of nanosheets. PIP diffuses into the surface through the gap between adjacent nanosheets and reacts with TMC forming a hollow structure above the nanosheets where the PIP concentration is lean. With more nanosheets at the interface, the gap density is higher, and its length is shortened. Consequently, the density and size of arched channels are changed. 0.135 wt% of g-$C_3N_4$ is found to give the most uniform structure with a hollow channel. Further increment in nanosheet content may lead to extensive stacking of nanosheets and disturb the ordering of structure. As revealed by the transmission electron microscopy (TEM) image of the membrane cross-section in Fig. 3f, the microscopically arched channels on the surface are hollow. By atomic force microscope (AFM) measurement, the surface area of the PIP/g-$C_3N_4$-TMC layer is found to be 1.76 times that of the PIP-TMC layer (see Supplementary Fig. 13). In the meantime, it is noticed that the thickness of PIP/g-$C_3N_4$ -PA membrane is $55 \pm 3$ nm, much lower than the control membrane ($83 \pm 5$ nm). Overall, the addition of g-$C_3N_4$ regulates polyamide formation both temporally and spatially, giving a thin, and nanoscale-ordered hollow structure on the surface.

The surface physiochemical properties were then characterized. XPS results in Fig. 3g–i and Supplementary Fig. 14 show that the characteristic N-C = N peak of g-$C_3N_4$ is absent from PA-g-$C_3N_4$ membrane, the N−H bond ratio scarcely changes (Fig. 3j, k), suggesting that g-$C_3N_4$ reside mostly at the bottom of the polyamide layer. It is also noticed the chemical composition and crosslinking degrees of both membrane surfaces are comparable, though PA-$C_3N_4$ has a slightly lower crosslinking degree (Supplementary Tables 4 and 5). Both PA and PA-g-$C_3N_4$ membranes are negatively charged at pH > 3, with the PA-g-$C_3N_4$ membrane being slightly more negative (see Supplementary Fig. 15). Noticeably, the isoelectric points are <pH 3, lower than most other NF membranes, likely due to the high TMC concentration of 0.4 wt% in the reactions that leaves abundant residual carboxylic groups on the surface. The water contact angle of the membrane is reduced to 24.1° (see Supplementary Fig. 16) due to its higher surface roughness. XRD patterns of the PA-g-$C_3N_4$ membrane give a d-spacing of 12.4 Å similar to that of the pure PA membrane (12.1 Å) (see Supplementary Fig. 17). The microstructure was also probed by positron annihilation spectroscopy (PAS). The $S$ parameter (Fig. 3l) quickly rises in PA-g-$C_3N_4$ until reaching a plateau, giving a lower selective layer thickness than that of the PA membrane. The slightly lower $S$ parameter of PA-g-$C_3N_4$ than the control membrane in the dense layer is likely due to the hindrance of positron diffusion by the g-$C_3N_4$ nanosheet.

## Fast permeation and precise ion-ion separation

In the NF test, the PA-g-$C_3N_4$ membrane exhibits water permeance of 105 L $m^{-2}$ $h^{-1}$ $bar^{-1}$, five times higher than that of the PA membrane, while at comparable $Na_2SO_4$ rejection of 99.4% (Fig. 4a). It outperforms state-of-the-art membranes in the permeance-rejection trade-off plot. To get more insights, we performed computational fluid dynamics (CFD) studies (see Supplementary Fig. 2). Figure 4b shows that more

transport pathways are found on the curvature of the nanoscale ordered surface leading to higher observed flux. It is consistent with the flow streamlines of smooth and nanoscale-ordered structures (Fig. 4c). However, if flux is normalized against the actual surface area, the intrinsic flux is, in fact, lower than that of smooth membranes. It could be related to the lower pressure drop rate near the surface of the arched structure (see Supplementary Figs. 24−27). Our analysis suggests that the nanoscale-ordered structure enhances water permeation via increased surface area. AFM characterization earlier reveals a ~1.76 times more surface area of the PA-g-$C_3N_4$ membrane, which partially explains its high permeance. In addition, the lower thickness resulting from the temporal and spatial effect of g-$C_3N_4$ also makes a substantial contribution.

Figure 4d shows that the MWCO of the PA-g-$C_3N_4$ membrane is 472 Da, which is slightly larger than that of the PA membrane. A narrow pore size distribution with a pore radius of 0.364 nm is achieved, which lies right between the sizes of hydrated $Cl^-$ ions (0.332 nm) and $SO_4^{2-}$ ions (0.379 nm) (Fig. 4e). Earlier, it is revealed that the membrane surfaces are highly negatively charged. The membrane is thus able to achieve high selectivity of 130 for $Cl^-$ over $SO_4^{2-}$ (Fig. 4f) through both size sieving and Donnan exclusion effects. The $Cl^-/SO_4^{2-}$ selectivity remains almost the same in the mixed salt tests with 2000 ppm of salts at varying $Cl^-$:$SO_4^{2-}$ weight ratios (Supplementary Fig. 18).

An additional benefit of our PA-g-$C_3N_4$ membrane is its self-cleaning property. A 100 ppm methylene blue (MB) solution was selected as the model sample to explore the photocatalytic performance under visible light. As shown in Fig. 4g, rejection remains steady at 98.4%, while flux keeps declining due to fouling, The flux can be recovered to 98.1% after three times recycling upon photocatalytic treatment. Supplementary Fig. 20 demonstrates that the PA-g-$C_3N_4$ membrane is much cleaner than the bare PA membrane after cleaning, and supplementary Figure 21 also signifies the important role of photocatalytic cleaning compared to soaking and rising. The mechanism of photocatalytic degradation of MB is illustrated in Supplementary Fig. 22. Electrons ($e^-$) and holes ($h^+$) are generated by visible light that partially penetrates through the polyamide layer to g-$C_3N_4$ [37], which then react with dissolved oxygen ($O_2$) to form superoxide radical anion $O_2^-$ [38]. MB is degraded by $h^+$ and $O_2^-$ to form $CO_2$ and $H_2O$, and thus membrane surface becomes clean again. Light is feasible for the PA-g-$C_3N_4$ membrane (see Supplementary Figure 23). This self-cleaning property shall simplify membrane cleaning operation, offering promises for practical application of the g-$C_3N_4$-based NF membrane in textile water purification and other applications where fouling is a concern. The separation performance of the PA-g-$C_3N_4$ membrane is stable for continuous operation (see Supplementary Fig. 19).

In summary, we have fabricated an ultra-permeable, high ion−ion selectivity and self-cleaning g-$C_3N_4$ hybridized NF membrane with a water flux of 105 L $m^{-2}$ $h^{-1}$ $bar^{-1}$ at a 99.4% rejection to $Na_2SO_4$, and a $Cl^-/SO_4^{2-}$ selectivity of 130. It represented one of the most competitive overall performances among the reported polymeric NF membranes. This study gives the strategy to tune the nanoscale ordered structure by g-$C_3N_4$ and reveals the role of such structure in transport. The technology offers opportunities for fast and precise separation.

## Methods
### General
Materials used in this work are provided in Chemicals and Materials in Supplementary Materials.

### Preparation of freestanding nanofilms
The 2.0 g PIP, 0.5 g trimethylamine (TEA), and 0.135 wt% g-$C_3N_4$ were dissolved in 100 mL water. Secondly, 0.16 mL phosphoric acid was added to adjust the solution pH to 9. Next, a 0.4 wt% TMC n-hexane solution was prepared. Then, the PIP solution was poured into a petri dish. After that, the TMC organic solution was gently added on top of

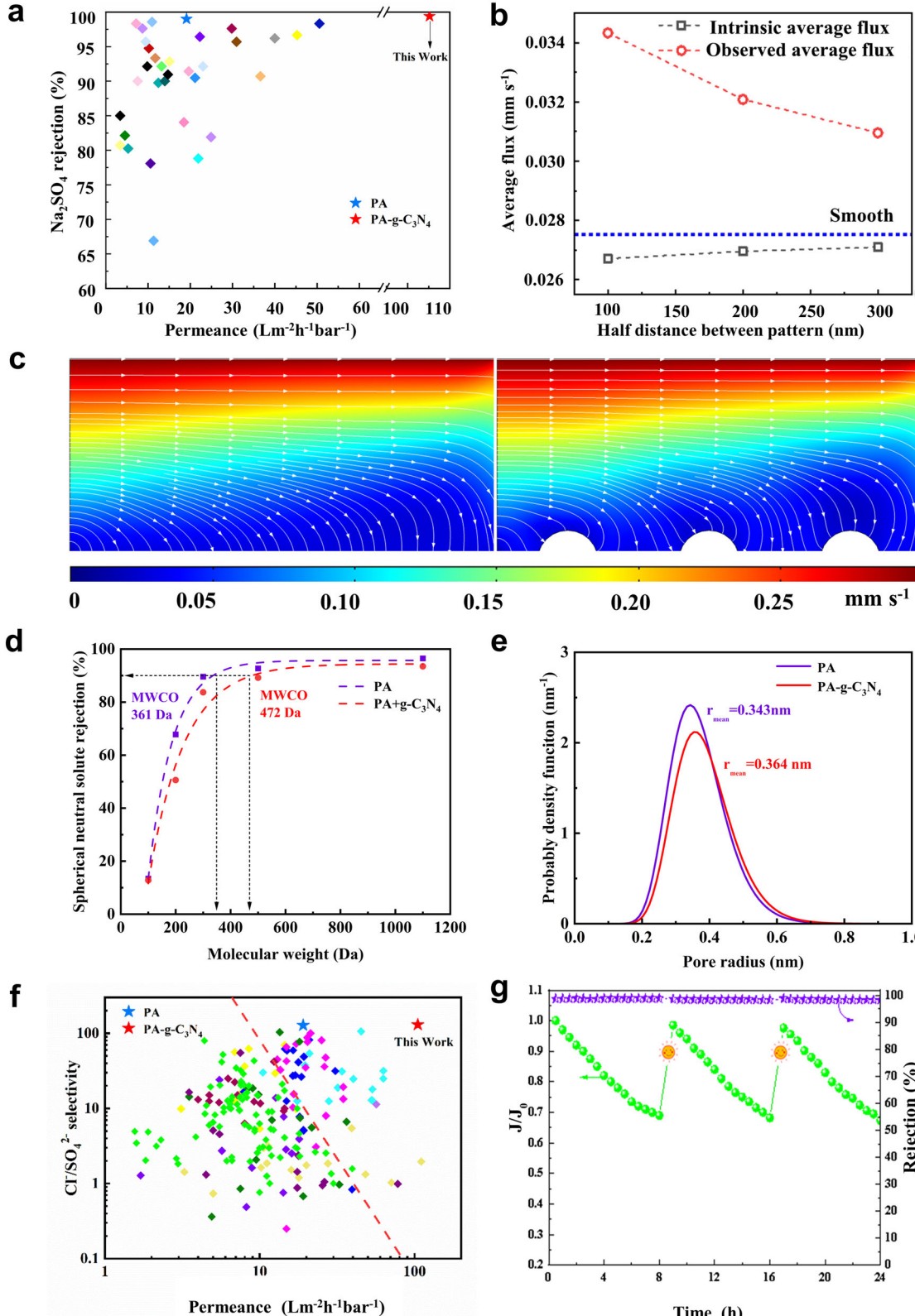

**Fig. 4 | Separation performances of NF membranes. a** Performance comparison of PA-g-C₃N₄ with state-of-the-art NF membranes. Literature data are obtained from ref. [3]. **b** Average flux of different half distances between patterns. **c** Comparison of the flow streamlines of smooth and nanoscale ordered structure. **d** MWCOs of the PA and the PA-g-C₃N₄ membrane. **e** Mean pore size of the PA and the PA-g-C₃N₄ membrane. **f** Trade-off between Cl⁻/SO₄²⁻ selectivity and water permeability of different NF membranes. Literature data are obtained from ref. [14]. **g** Flux decline and rejection of the PA-g-C₃N₄ membrane with methylene blue as the feed solute; cleaning was done with visible light. Source data are provided as a Source Data file.

the aqueous solution to induce IP. After 5 min, a freestanding film was formed that was taken out of the petri dish, washed with DI water three times, and stored in DI water.

## Preparation of PA and PA-g-C₃N₄ membranes

The PA and PA-g-C₃N₄ NF composite membranes were prepared by IP on top of the polyether sulfone (PES, MWCO 100 kDa) ultrafiltration (UF) membranes. The aqueous solution was prepared with 0.2–2.0 g of PIP and 0.5 mL TEA in 100 mL water, and its pH was tuned to 9 by adding 0.16 mL phosphoric acid. To prepare the PA-g-C₃N₄ composite membrane, g-C₃N₄ nanosheet was added at the concentrations of 0.075 to 0.145 wt% to form well-dispersed suspensions. The organic solution was prepared by dissolving 0.1–0.5 wt% TMC in n-hexane. Then, the PES membrane was dipped into the aqueous phase for 5 min. After being taken out, the excess solution on its surface was removed using a rubbery roller, and the support was left at room temperature until the surface appeared dull and dry. Next, the membrane was immersed in the organic phase for 1 min. After that, the membrane was heated in an oven at 80 °C for 5 min, and stored in DI water before use. Three replicate membranes for each experiment were fabricated and examined. Error bars represent the standard deviation of the three membranes.

## Characterization

SEM images were taken by a Hitachi S4800 cold field emission scanning electron microscopy. TEM images were conducted by an FEI Tecnai G2 F20 S-TWIN 200KV field-emission transmission electron microscopy. The elemental composition of the membrane surface was analyzed by an X-ray photoelectron spectrometer (XPS, Thermo ESCALAB250Xi, USA). XRD was tested by a Bruker AXS D8 Advance powder X-ray diffractometer. AFM images were measured by NanoScope MultiMode scanning probe microscopy (Veeco, Camarillo, California, US).

## Membrane performance test

The permeation performance of the membrane was measured on the cross-flow filtration equipment. The effective area of the membrane was 26 cm². The solution temperature was maintained at 25 °C by a heat exchanger. In order to achieve a steady state, the membranes were pre-pressurized for 2 hours under 6 bar. The flow rate was 1.5 L min⁻¹. The concentrations of salts (including single NaCl and Na₂SO₄) and dyes in feed solutions were 2000 ppm and 100 ppm, respectively.

The water permeance was calculated based on Eq. 1.

$$L = \frac{V}{A \cdot \Delta t \cdot \triangle p} \tag{1}$$

where $L$ is the water permeance (L m⁻² h⁻¹ bar⁻¹), $V$ (m³) is the volume of permeate collected over $\Delta t$, $A$ is the effective membrane area (m²), $\Delta t$ and $\Delta p$ represent the filtration time (h), and the transmembrane pressure.

$$R = \left(1 - \frac{C_p}{C_f}\right) \times 100\% \tag{2}$$

where $R$ is the salt rejection (%), $C_p$ and $C_f$ represent the concentration of permeate and feed solutions, respectively. The salt concentration was quantified by conductivity measurement, and dye solutions were measured by UV−vis.

In addition, mixed salt tests were performed in a similar way using the mixed solutions of NaCl and Na₂SO₄, where the total concentrations were kept constant at 2000 ppm while the weight ratios of Cl⁻:SO₄²⁻ were varied. The Cl⁻ and SO₄²⁻ concentrations were measured by ion chromatography.

The Cl⁻/SO₄²⁻ selectivity was calculated based on the following equation:

$$S_{Cl^-/SO_4^{2-}} = \frac{1 - R_{Cl^-}}{1 - R_{SO_4^{2-}}} \tag{3}$$

## MD simulation

MD simulations in this work were conducted using the large-scale atomic/molecular massively parallel simulator (LAMMPS)[39] package on a parallel Linux cluster. The results were visualized using the VMD[40], and the initial setup was constructed with the Materials Studio 8.0 software. Water molecules were modeled using the SPC potential[41], whereas other molecules were characterized using the OPLS−AA force field[42], the parameters for which are shown in Supplementary Table 1. For all dynamics runs, the temperature was controlled using the Nosé−Hoover thermostat, and the pressure was controlled with the Berendsen barostat. The Lennard-Jones (LJ) and coulomb potentials were combined to compute pair interactions between atoms. The particle−particle−mesh (PPPM) k-space solver[43] was used to address the long-range columbic interactions with a relative accuracy of 10⁻⁶. Two computing models were constructed in this work. Model A in Fig. 2a composes of an n-hexane phase and a water phase containing PIP molecules. The number of molecules and the cell dimensions can be found in Supplementary Table 2. A comparative system was likewise constructed with the addition of g-C₃N₄. The densities of water and n-hexane, as well as the concentrations of PIP, were determined in accordance with ref. 44. Throughout the simulation, the SHAKE algorithm[45] was used to constrain the bond and angle of water molecules. Initially, energy minimization was utilized to eliminate atom overlaps. Following that, a 10 ns NVT run ($T = 300$ K) was performed to obtain data for further analysis. The non-bonded interaction energy between g-C₃N₄ and the other species was determined using Model B. A g-C₃N₄ layer was put at the bottom of the simulation cell, and water, n-hexane, and PIP molecules were deposited on it independently in each of the three simulation cells; the interaction energy between them and the g-C₃N₄ layer was determined after equilibrium had been attained.

## Data availability

The authors declare that the data supporting the findings of this study are available within the paper and its supplementary information file. Source data are provided in this paper. All data are also available by request to the corresponding authors. Source data are provided in this paper.

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

## Acknowledgements

This work was financially supported by the National Natural Science Foundation of China (21878323), the Fund for Strengthening Technical Fields of Basic Plan (2021-JCJQ-JJ-0128), the National Key Research and Development Program of China (2020YFC1807901), the Scientific Research Foundation of China Agricultural University (2021RC022), Ministry of Education of Singapore Tier 1 Grant (A-8000192-01-00).

## Author contributions

C.Z. and S.Z. conceived the idea, designed the experiments, and supervised the project. C.Z., Y.Z., B.L., W.T., and R.M. performed experiments, including materials synthesis, membrane preparation, characterization analyses, and membrane separation performance measurements, and drew the figures. Y.J. performed MD studies, and C.S. conducted CFD simulations. P.L. and S.L. contributed to discussions and data interpretation. All authors contributed to the drafting and revision of the paper.

## Competing interests

The authors declare no competing interests.

## Additional information

**Peer review information** : *Nature Communications* thanks the anonymous reviewer(s) for their contribution to the peer review of this work. Peer reviewer reports are available.

