## [Peer Review File · Nature Communications]

Polyamide membranes with nanoscale ordered structures for fast permeation and highly selective ion-ion separationREVIEWER COMMENTS

Reviewer #1 (Remarks to the Author):

This paper reported the fabrication of ultrahigh flux NF membrane by incorporation of g-C₃N₄ in PIP aqueous solution during interfacial polymerization. Comparing with other method reported, this paper did show major advantage in flux increase, as shown in trade-off curve. Therefore I recommend for publication with addressing following problems:

1. When author check the stability test, why using methylene blue dye instead of salt?
2. In the preparation of free-standing nanofilm, TMC concentration (0.4%) used for IP interfacial polymerization is abnormally high, why? As normally TMC concentration is used less than 0.2%
3. In the experimental section for interfacial polymerization, with g-C₃N₄ case, what the concentration for each reagents and additives? Please give it in detail
4. What is the chemical nature of PIP/ g-C₃N₄ interactions, also the nature of hexane/g-C₃N₄ as mentioned in the paper.
5. How is the hollow strand morphology formed? Please using the data to explain the chemical nature and kinetics.

Reviewer #2 (Remarks to the Author):

This manuscript describes the addition of graphitic carbon nitride (g-C₃N₄) to interfacial polymerization solutions to enhance the permeability of nanofiltration membranes. Many papers have examined the effects of nanoparticles on interfacial polymerization, but this work shows uniquely high water permeabilities along with nearly complete Na₂SO₄ rejection. The authors propose that high permeability is due to a low membrane thickness and high surface area that result from g-C₃N₄ limiting piperazine diffusion to specific regions during interfacial polymerization. The permeation rate combined with the high Na₂SO₄ rejection is remarkable. Thus, I recommend publication after some significant changes. These include clearly specifying the interfacial polymerization procedure so others can repeat the work, discussing whether the technique is reproducible for multiple batches of membranes, and clarifying parts of the manuscript. Specific comments follow.

1. The authors should note what the "g" stands for in "g-C₃N₄".
2. The manuscript mentions Cl⁻/SO₄²⁻ selectivity in the introduction, but I never saw any data on chloride permeation. Did the authors do an experiment with NaCl rejection or some mixed-salt experiment? It is not clear how they determined Cl⁻/SO₄²⁻ selectivity. Was the NaCl rejection around 20% as the selectivity values suggest? Do the authors know why the control and g-C₃N₄ membranes show such high selectivity compared to most NF membranes?
3. The resolution in some figures is not high enough.
4. The authors should probably reduce the number of significant figures in the permeability value 105.2. The 0.2 is not significant. Also, the manuscript should state how many replicate membranes were examined and if the reproducibility holds for several different batches of interfacial polymerization. This is vital to demonstrate.
5. I am not sure why the manuscript refers to piperazine as an activator. The trimesoyl chloride is already activated, and piperazine is a reactant. Also, what is the inhibitor to which they refer. I looked up reference 29 but did not see this terminology.
6. The authors discuss that g-C₃N₄ slows piperazine diffusion through interactions with g-C₃N₄. What is the ratio of g-C₃N₄ to piperazine in the solution? Is most of the piperazine adsorbed on g-C₃N₄ or is the effect of g-C₃N₄ primarily at the interface?
7. I don't understand Figures 2g and 2h. In Figure 2g, what are the meanings of the colors. In Figure 2h, the two plots do not look that different, and I am not sure of the meaning of the x-axis.
8. The TEM images in Figure 3 are difficult to compare.
9. Figures 4a, 4d, and 4e need labels on the axes. I am not sure how the authors are determining the pore-size distribution in Figure 4e.
10. The photocatalytic cleaning raises questions. How will you get light into a membrane module? Are reactive species diffusing through the membrane from the underlying g-C₃N₄ to react with methylene blue? Did the authors perform any control experiments to verify that light is required

for the membrane regeneration? Does the same procedure without light but with rinsing give a similar result? I am not sure that this part of the manuscript is necessary, but if included it should contain control experiments.

11. The procedure for membrane preparation does not give the amounts of materials dissolved in solutions. Are concentrations the same as for the free-standing film? The free-standing film does not mention phosphoric acid, but membrane fabrication does. The reader needs sufficient detail to reproduce the procedure. What was the effective area of the membranes in the permeation tests? How long did the permeation occur to achieve steady state?

12. Can the authors give uncertainties in Table S6 and note how many different membranes were used in each case? These and other experiments should employ replicate membranes, not just multiple experiments on the same membranes.

13. If concentration polarization is not large, the membrane rejection will increase with flow rate. Is this a possible reason why the membranes show such high Na₂SO₄ rejection along with high permeability?

14. The manuscript suggests that slow spreading "confirms that the diffusion rate of the (PIP+g-96 C3N4)/TMC system is slower than that of the PIP/TMC system." Spreading and diffusion are not the same. Perhaps the term mixing, rather than diffusion, is appropriate. This seems more of a mixing and spreading phenomenon.

Reviewer #3 (Remarks to the Author):

Present work demonstrates fast permeation and selective ion separation by regulation of interfacial polymerization (IP). g-C₃N₄ is applied for controlled diffusion of PIP by one order of magnitude, thereby forming a nanoscale-ordered hollow cone structure. MD simulation explains the role of g-C₃N₄ in the IP process to understand the role of membrane microstructure in the nanofiltration process. Although I cannot recommend this manuscript for publication in Nature Communications due to lack of novelty, let me advise the following points for the authors' consideration.

1. Incomplete literature survey: The authors need to survey previously reported papers such as "Synthesis and characterization of g-C₃N₄ nanosheet modified polyamide nanofiltration membranes with good permeation and antifouling properties" and "g-C₃N₄ nanofibers network reinforced polyamide nanofiltration membrane for fast desalination". I suggest mentioning of these works in the introduction section with emphasizing the novelty of the present work in their respect.
2. The section titled "Ultrafast permeation and ion-ion separation" requires more discussion. For example, the authors studied separation performance of PA-g-C₃N₄ as a function of g-C₃N₄ concentration and failed to explain the reason of decreased flux beyond 0.145 % of g-C₃N₄.
3. In page 9, line3, the authors mention that the Cl⁻/SO₄²⁻ selectivity reaches 130 without a technical detail. Experiments carried out for the selectivity measurement need be provided.
4. The authors presented the self-cleaning property of PA-g-C₃N₄ as one of the highlights of the work whereas they do not demonstrate or explain the mechanism.
5. Self-cleaning properties of PA and PA-g-C₃N₄ need be compared and emphasized. At the same time, the authors may want to consider visual demonstration of the anti-fouling behaviour of PA and PA-g-C₃N₄ via SEM images of PA and PA-g-C₃N₄ before and after the fouling tests.

Reply to Comments of Reviewers

Responses to the Reviewer #1:

This paper reported the fabrication of ultrahigh flux NF membrane by incorporation of g-C₃N₄ in PIP aqueous solution during interfacial polymerization. Comparing with other method reported, this paper did show major advantage in flux increase, as shown in trade-off curve. Therefore I recommend for publication with addressing following problems.

Response:

Thank you for the insightful comments on our paper and helpful suggestions.

Comments 1:

When author check the stability test, why using methylene blue dye instead of salt?

Response:

Thank you for the comment. We used methylene blue dye solution to carry out the long term test was to demonstrate the photo-catalytic cleaning property of our membrane. The fouling behavior of methylene blue is more obvious than salt to investigate membrane performance. We have also done the stability test with a NaSO₄ solution. As shown in the Supplementary Figure 19 (which was also given below), in a 72h test salt rejection and water flux were barely changed. It showed that the membrane was stable for use.

Figure S19. The stability test of permeance and Na₂SO₄ rejection for PA-g-C₃N₄ membrane.

Comments 2:

In the preparation of free-standing nanofilm, TMC concentration (0.4%) used for IP interfacial polymerization is abnormally high, why? As normally TMC concentration is used less than 0.2%.

Response:

Thank you for the comment. We agree that normally TMC concentration is less than 0.2 wt%. In this study, we studied different TMC concentrations at 0.135 wt% g-C₃N₄ to investigate their effects on the interfacial polymerization. The surface morphologies are given below. It can be seen that with the increase of TMC concentration till 0.4 wt%, the nanoscale structure becomes more ordered which increases the surface area and roughness. When TMC concentration reaches 0.5 wt%, the surface becomes less regular, likely due to the extensive crosslinking that limit the formation of nanoscale hollow structure. And the permeance of this membrane at 0.4 wt% was the highest among all 5 concentrations (see Supplementary Table 6) under high Na₂SO₄ rejection of 99%. The corresponding results have been added into the Supplementary Figure 11 and Supplementary Table 6 in the revised supporting information.

Figure S11. SEM surface morphologies of different TMC concentrations. (a) 0.1 wt%. (b) 0.2 wt%. (c) 0.3 wt%. (d) 0.4 wt%. (e) 0.5 wt%.

Supplementary Table 6. The permeance of the membranes with different TMC concentrations.

Different TMC concentration (wt%)	Permeance (L m ⁻² ·h ⁻¹ ·bar ⁻¹)
0.1	32 ± 1

0.2	56 ± 1
0.3	73 ± 2
0.4	103 ± 2
0.5	75 ± 1

Comments 3:

In the experimental section for interfacial polymerization, with g-C₃N₄ case, what the concentration for each reagents and additives? Please give it in detail.

Response:

Thank you for the comment. We have provided the experimental details in the Methods session of the manuscript. Revisions are as follows:

Preparation of freestanding nanofilms. Firstly, 2.0g PIP, 0.5g trimethylamine (TEA) and 0.135 wt% g-C₃N₄ were dissolved in 100 mL water. Secondly, 0.16 mL phosphoric acid was added to adjust the solution pH to 9. Next, a 0.4 wt% TMC n-hexane solution was prepared. Then, the PIP solution was poured into a petri dish. After that, the TMC organic solution was gently added on top of the aqueous solution to induce interfacial polymerization. After 5 min, a freestanding film was formed that was taken out of the petri dish, washed by DI water for three times and stored in DI water.

Preparation of PA and PA-g-C₃N₄ membranes. The PA and PA-g-C₃N₄ NF composite membranes were prepared by IP on top of the polyether sulfone (PES, MWCO 100 kDa) ultrafiltration (UF) membranes. The aqueous solution was prepared with 0.2 to 2.0 g of PIP and 0.5mL TEA in 100 mL water, and its pH was tuned to 9 by adding 0.16 mL phosphoric acid. To prepare the PA-g-C₃N₄ composite membrane, g-C₃N₄ nanosheets were added at the concentrations of 0.075 to 0.145 wt%. The organic solution was prepared by dissolving 0.1 to 0.5 wt% TMC in n-hexane. Then, the PES membrane was dipped into the aqueous phase for 5 min. After being taken out, the excess solution on its surface was removed using a rubbery roller and the support was left at room temperature until the surface appeared dull and dry. Next, the membrane was immersed in the organic phase for 1 min. After that, the membrane was heated in an oven at 80 °C for 5 min, and stored in DI water before use. Three replicate membranes for each experiment were fabricated and examined to get a reproducible performance.

Membrane performance test. The permeation performance of the membrane was measured on the cross-flow filtration equipment. The effective area of the membrane was 26 cm². The solution temperature was maintained at 25°C by a heat exchanger. In order to achieve steady state, the membranes were pre-pressurized for 2 hours under 6 bar. The flow rate was 1.5 L·min⁻¹. The concentrations of salts (including single NaCl, Na₂SO₄) and dyes in feed solutions were 2000 ppm and 100 ppm, respectively.

The water permeance was calculated based on Eq. 1 .

$$L = \frac{V}{A \cdot \Delta t \cdot \Delta p} \quad (1)$$

where L is the water permeance ($\text{L m}^{-2} \text{ h}^{-1} \text{ bar}^{-1}$), V (m^3) is the volume of permeate collected over Δt , A is the effective membrane area (m^2), Δt and Δp represent the filtration time (h) and the transmembrane pressure (bar).

$$R = \left(1 - \frac{C_p}{C_f}\right) \times 100\% \quad (2)$$

where R is the salt rejection (%), C_p and C_f represent the concentration of permeate and feed solutions, respectively. The salt concentration was quantified by conductivity measurement, and dye solutions were measured by UV-Vis.

In addition, mixed salt tests were performed in a similar way using the mixed solutions of NaCl and Na_2SO_4 , where the total concentrations were kept constant at 2000 ppm while the weight ratios of $\text{Cl}^-:\text{SO}_4^{2-}$ were varied. The Cl^- and SO_4^{2-} concentrations were measured by ion chromatography.

The $\text{Cl}^-/\text{SO}_4^{2-}$ selectivity was calculated based on the following equation:

$$S_{\text{Cl}^-/\text{SO}_4^{2-}} = \frac{1 - R_{\text{Cl}^-}}{1 - R_{\text{SO}_4^{2-}}} \quad (3)$$

Comments 4:

What is the chemical nature of PIP/ g- C_3N_4 interactions, also the nature of hexane/g- C_3N_4 as mentioned in the paper.

Response:

Thank you for the comment. We investigated the nature of PIP/ g- C_3N_4 and hexane/g- C_3N_4 interactions by MD simulations. We used the LAMMPS package to calculate the interaction energy, including Van der Waals force, hydrogen bonding and electrostatic interaction energy between g- C_3N_4 and other molecules. The results in Fig. 2c provides an overall comparison of such interactions between g- C_3N_4 and water, hexane and PIP. To gain further sights, The radial distribution functions (RDF, $g(r)$) of one representative atom in g- C_3N_4 , named as N1, around different atoms in PIP, water, and n-hexane at the start (Fig. 2d) and end (Fig. 2e) of the simulation were computed and given in Fig. 2d. It is noticed that the peak height for N1- H_2O is reduced and N1 becomes closely surrounded by the nitrogen atoms in PIP at the end of simulation, likely through hydrogen bonding. A similar trend is observed with C atom in g- C_3N_4 (see Supplementary Figure 10). It suggests that the hydrogen bonding between g- C_3N_4 and PIP may contribute to strong interactions. We have included such discussions in the manuscript.

For the interaction between g-C₃N₄ and hexane, the nitrogen and carbon in g-C₃N₄ is found to interact closely with the carbon in hexane (Fig. 2d). It is difficult to determine the exact nature of interactions using the current methods, but based on our understanding, we think it's van der Waals force.

Comments 5:

How is the hollow strand morphology formed? Please using the data to explain the chemical nature and kinetics.

Response:

Thank you for your comment. In this work, based on MD simulations, we provided explanations on the hollow strand morphology in the revised manuscript as follows:

“The emergence of ordered structure might be attributed to the retarded diffusion of PIP in the presence of g-C₃N₄, similar to the local activation and lateral inhibition phenomenon reported in earlier studies^{23,35,36}. However, it is interesting that the number of arched channels increases and channel size decreases as g-C₃N₄ content rises until reaching 0.135 wt%. This is well explained by the tiling effect of nanosheets. PIP diffuses into the hexane phase through the gap between adjacent nanosheets, reacts with TMC at the other side, forming a hollow structure above the nanosheets where the PIP concentration is lean. With more nanosheets at the interface, the gap density is higher and its length is shortened. Consequently, the density and size of arched channels are changed. 0.135 wt% of g-C₃N₄ is found to give the most uniform structure with a hollow channel. Further increment in nanosheet content may lead to extensive stacking of nanosheets and disturb the ordering of structure.”

Responses to the Reviewer #2:

This manuscript describes the addition of graphitic carbon nitride (g-C₃N₄) to interfacial polymerization solutions to enhance the permeability of nanofiltration membranes. Many papers have examined the effects of nanoparticles on interfacial polymerization, but this work shows uniquely high water permeabilities along with nearly complete Na₂SO₄ rejection. The authors propose that high permeability is due to a low membrane thickness and high surface area that result from g-C₃N₄ limiting piperazine diffusion to specific regions during interfacial polymerization. The permeation rate combined with the high Na₂SO₄ rejection is remarkable. Thus, I recommend publication after some significant changes. These include clearly specifying the interfacial polymerization procedure so others can repeat the work, discussing whether the technique is reproducible for multiple batches of membranes, and clarifying parts of the manuscript. Specific comments follow.

Response:

Thank you for your positive evaluation of our work and the insightful comments. Following your suggestions, we have provided details of the interfacial polymerization process so that others can repeat the work and revised our manuscript for better clarity.

Comments 1:

The authors should note what the “g” stands for in “g-C₃N₄”.

Response:

Thank you for the comment. We are sorry for forgetting to note the “g” stands for in “g-C₃N₄”. The “g” for in “g-C₃N₄” has been noted in the revised manuscript as follow:

“graphitic carbon nitride (g-C₃N₄) nanosheet.”

Comments 2:

The manuscript mentions Cl⁻/SO₄²⁻ selectivity in the introduction, but I never saw any data on chloride permeation. Did the authors do an experiment with NaCl rejection or some mixed-salt experiment? It is not clear how they determined Cl⁻/SO₄²⁻ selectivity. Was the NaCl rejection around 20% as the selectivity values suggest? Do the authors know why the control and g-C₃N₄ membranes show such high selectivity compared to most NF membranes?

Response:

Thank you for the comment. We did experiment with NaCl rejection and some mixed-salt experiment and measured the rejection to Cl^- and SO_4^{2-} in the revised manuscript. Mixed salt tests were performed using the mixed solutions of NaCl and Na_2SO_4 , the weight ratios of $\text{Cl}^-:\text{SO}_4^{2-}$ in 2000 ppm mixed solution were varied. The Cl^- and SO_4^{2-} concentrations were measured by ion chromatography. The $\text{Cl}^-/\text{SO}_4^{2-}$ selectivity was calculated based on Eq. (3). Details of the experiment were given in the Methods session of the revised manuscript.

The results from mixed salt test were provided in Supplementary Figure 18. The Cl^- rejection is around 20%, and the rejection to SO_4^{2-} is $> 99\%$ in all tests. The $\text{Cl}^-/\text{SO}_4^{2-}$ selectivity is 130. The high selectivity is attributed to the negative surface charge on membranes and the suitable pore size. Fig. 4f compares our membranes with other membranes reported in literature. In fact, a few other membranes can also reach such high selectivity, but their permeance is much lower than our PA-g- C_3N_4 membrane. The highlight of our work is boosting the membrane permeance while maintaining high selectivity.

Figure S18. Separation performance of the weight ratio of $\text{Cl}^-/\text{SO}_4^{2-}$ in 2000 ppm mixed solution. (a) Rejection. (b) $\text{Cl}^-/\text{SO}_4^{2-}$ selectivity.

We explained in the manuscript as follows:

“Fig. 4d shows that the MWCO of PA-g- C_3N_4 membrane is 472 Da, which is slightly larger than that of PA membrane and is consistent with earlier discussions. A narrow pore size distribution with pore radius of 0.364 nm is achieved, which lies right between the sizes of hydrated Cl^- ions (0.332 nm) and SO_4^{2-} ions (0.379 nm) (Fig. 4e). It is also revealed that the membrane surfaces are highly negatively charged. The membrane is thus able to achieve high selectivity of 130 for Cl^- over SO_4^{2-} (Fig. 4f) through both size sieving and Donnan exclusion effects. The $\text{Cl}^-/\text{SO}_4^{2-}$ selectivity remains

almost the same in the mixed salt test with 2000 ppm mixed salt solution varying Cl⁻:SO₄²⁻ weight ratios (Supplementary Figure 18).”

Comments 3:

The resolution in some figures is not high enough.

Response:

Thank you for the comment. We apologize for the problem with figure quality, and has now increased the resolutions of our figures in the revised manuscript.

Comments 4:

The authors should probably reduce the number of significant figures in the permeability value 105.2. The 0.2 is not significant. Also, the manuscript should state how many replicate membranes were examined and if the reproducibility holds for several different batches of interfacial polymerization. This is vital to demonstrate.

Response:

Many thanks for your kind comments. According to your suggestion, we have reduced the number of significant figures in the permeability value 105.2 to 105. Also, three replicate membranes for each experiment were examined and the data were consistent.

The details have been added into the Methods session of the revised manuscript as follow:

“Three replicate membranes for each experiment were fabricated and examined to get a reproducible performance.”

Comments 5:

I am not sure why the manuscript refers to piperazine as an activator. The trimesoyl chloride is already activated, and piperazine is a reactant. Also, what is the inhibitor to which they refer. I looked up reference 29 but did not see this terminology.

Response:

Thank you for your comment. We used the terminology of activator and inhibitor based on other studies in literature; and we apologize for the wrong reference used, which should be reference 23 instead. However, we do agree that the concepts here are blur and hence removed relevant discussions in the revised manuscript. The discussions are revised as ‘When g-C₃N₄ was added into the PIP solution, the monomer diffusivity decreases to 10⁻¹¹ m² s⁻¹, one order of magnitude lower than its original value. By real-time online optical monitoring (see Supplementary Figure 8), it is observed that the spreading area of the (PIP+g-C₃N₄)/TMC system (Fig.1b) is smaller than that of the PIP/TMC system within the same time (Fig.1c), which confirms that the mixing of spreading and diffusion within the (PIP+g-C₃N₄)/TMC system is slower than that with the PIP/TMC system. Hence, the diffusion rates of the amino monomers may be adjusted by tuning the g-C₃N₄ to reach an ideal difference in the diffusion of monomers in both the aqueous and organic phases’.

Comments 6:

The authors discuss that g-C₃N₄ slows piperazine diffusion through interactions with g-C₃N₄. What is the ratio of g-C₃N₄ to piperazine in the solution? Is most of the piperazine adsorbed on g-C₃N₄ or is the effect of g-C₃N₄ primarily at the interface?

Response:

Thank you for the question. The mass ratio of g-C₃N₄ to piperazine in the water solution is 1:2.5. As is mentioned in your question, based on our MD simulations, the slowed diffusion of PIP in water might be due to two effects: 1) The interaction (or adsorption) of PIP with g-C₃N₄; since the quantity of g-C₃N₄ in water is 2.5 times less (and roughly 2.5 times less in molar ratio) than PIP, the majority of PIP molecules are free. However, even if only part of the PIP are adsorbed onto g-C₃N₄, they form large complex in water, increase the viscosity of the solution, and may then hinder the diffusion of other PIP molecules through the spatial and viscous effects; hence, the hindrance effect caused by PIP adsorption onto g-C₃N₄ could be substantial; 2) Nanosheets tend to tile at the interface, which causes direct spatial barrier for PIP diffusion. It is difficult to quantify the relative importance of each effect, as MD only gives qualitative explanations. We tend to believe that both effects play important roles with tiling effects being even more substantial given the low concentration of g-C₃N₄ in the solution.

Comments 7:

I don't understand Figures 2g and 2h. In Figure 2g, what are the meanings of the colors. In Figure 2h, the two plots do not look that different, and I am not sure of the meaning of the x-axis.

Response:

Thank you for your question. In Fig 2g, different colors indicate the diffusion path of four different pieces of g-C₃N₄ constructed in the MD simulation system. This figure tells us that all the four nanosheets migrate to the interface and tile on it. In Fig 2h, the x-axis indicates the spatial position on z-axis matching the Fig 2a. The PIP molecules are more intent to migrate to the interface of the two phases with the absence of g-C₃N₄. On the other hand, the diffusion is more likely to be prohibited and the interface plane would be closer to the water phase (to the right-hand side) when g-C₃N₄ is present. We have clarified the information in the caption of Figs. 2h and 2g as “Time dependence of the position of four pieces of g-C₃N₄ nanosheets along z-axis; and h the PIP number density along z axis in the simulation systems. Z-axis starts from the hexane phase to the water phase as is captured in 2a.”

Comments 8:

The TEM images in Figure 3 are difficult to compare.

Response:

Many thanks for your kind comments. We have edited the TEM images in Figure 3 in the revised manuscript.

The corresponding TEM images in Figure 3 is given below:

Fig. 3 Characterizations of NF membranes. **a** The SEM surface morphology of the PA membrane. **b** The AFM image of the PA membrane. **c** The TEM morphology of PA membrane. **d** The SEM surface morphology of the PA-g-C₃N₄ membrane. **e** The AFM image of the PA-g-C₃N₄ membrane; **f** The TEM morphology of PA-g-C₃N₄ membrane. **g-i** C1s XPS spectra for g-C₃N₄, PA, and PA-g-C₃N₄ membrane respectively. **j, k** N1s XPS spectra for PA and PA-g-C₃N₄ membrane respectively. **l** S parameter as a function of positron energy for PA and PA-g-C₃N₄ membrane. The concentration of g-C₃N₄ for membrane preparation was 0.135 wt%.

Comments 9:

Figures 4a, 4d, and 4e need labels on the axes. I am not sure how the authors are determining the pore-size distribution in Figure 4e.

Response:

Many thanks for your kind comments. We are sorry for the missing labels. We have now added the labels on the axes of Figs. 4a, 4d and 4e in the revised manuscript and given below:

Fig. 4 a Performance comparison of PA-g-C₃N₄ with state-of-the-art NF membranes. b Average flux of different half distance between pattern. c Comparison the flow streamlines of smooth and nanoscale ordered structure. d MWCOs of the PA and the PA-g-C₃N₄ membrane. e Mean pore size of the PA and the PA-g-C₃N₄ membrane. f Trade-off between Cl⁻/SO₄²⁻ selectivity and water permeability of different NF membranes from ref 14. g Flux decline and rejection of the PA-g-C₃N₄ membrane with methylene blue as the feed solute and simulated visible light.

In addition, we have added the methods for measurement of pore-size distribution in section 3.1 of the revised supporting information.

The corresponding descriptions are also given below:

“By calculating the Stokes radius of the spherical solute of interest and neglecting the effect of steric and hydrodynamic interactions between solute and pore space on solute rejection, the pore size distribution of the NF membrane can be expressed as the following probability density function.

$$\frac{dR(r_p)}{dr_p} = \frac{1}{r_p \ln\sigma_p \sqrt{2\pi}} \exp\left[-\frac{(\ln r_p - \ln\mu_p)^2}{2(\ln\sigma_p)^2}\right] \quad (S4)$$

where μ_p is the average pore size of the composite membrane, μ_p is the Stokes radius of the spherical solute corresponding to a rejection rate of 50.0%, σ_p is the ratio between the Stokes radius with a rejection of 84.13% to the Stokes radius with a rejection of 50.0%, r_p is the Stokes radius of the solutes.”

Comments 10:

The photocatalytic cleaning raises questions. How will you get light into a membrane module? Are reactive species diffusing through the membrane from the underlying g-C₃N₄ to react with methylene blue? Did the authors perform any control experiments to verify that light is required for the membrane regeneration? Does the same procedure without light but with rinsing give a similar result? I am not sure that this part of the manuscript is necessary, but if included it should contain control experiments.

Response:

Many thanks for your comments. We have done additional experiment to answer the questions and further confirm our explanations:

1) In our experiment, after the membrane was removed from the membrane module, light was used to irradiate the membrane to perform the photocatalytic cleaning. It is a good question how it could be possible to get light into a large membrane module in the real applications. Most flat sheet membrane modules for NF today are spiral wound and it is difficult to disassemble the module for cleaning. In this case, we may adopt a different module configuration, such as plat and frame, which is convenient for photocatalytic cleaning. The photocatalytic cleaning might be cost effective and

environmental friendly in the future.

2) For your second question, are reactive species diffusing through the membrane from the underlying g-C₃N₄ to react with methylene blue? Yes, we think reactive species diffuse through the membrane from the underlying g-C₃N₄ to react with methylene blue. The reaction between g-C₃N₄ and organic species has been studied in literature and the mechanism is illustrated in Supplementary Figure 22. In order to verify the occurrence of reaction on our membrane, we have done additional experiment. Specifically, PA membrane and PA-g-C₃N₄ membrane were immersed in methylene blue solution (100 ppm, 50 ml) and irradiated under light for 180 min. In addition, the PA-g-C₃N₄ membrane in methylene blue solution (100 ppm, 50 mL) under dark conditions was also studied as a control. The UV-Vis absorbance of the solution was measured at the start and end of the experiment.

The corresponding results were shown below in Supplementary Figure 23. The absorbance barely changes for the PA membrane. On the other hand, the absorbance decreases substantially with the PA-g-C₃N₄ membrane (Supplementary Figure 23b). Without light, no change is observed (Supplementary Figure 23c). Hence, g-C₃N₄ plays a major role for the photocatalytic decomposition of MB. In addition, the FTIR spectra of the PA-g-C₃N₄ membrane shows no change before and after irradiation (Supplementary Figure 23d).

Figure S23. The UV-Vis spectra of the membranes after immersion in the methylene blue solution and then exposed to light for different durations. (a) PA membrane with exposure to light; (b) PA-g-C₃N₄ membrane with exposure to light. (c) PA-g-C₃N₄ membrane kept in the dark. (d) The FTIR of PA-g-C₃N₄ membrane before and after exposure to light.

3) For your third question, did the authors perform any control experiments to verify that light is required for the membrane regeneration? Does the same procedure without light but with rinsing give a similar result? Based on your suggestions, we have conducted control experiment to verify the role of light. Specifically, we applied different cleaning methods to the fouled PA-g-C₃N₄ membrane, including soaking, rinsing without light, and rinsing with light. The SEM images of membrane surfaces after cleaning as captured below show that foulants were removed more efficiently with light irradiation than the other two cleaning methods. The existence of light is crucial for membrane cleaning.

Figure S21. The SEM images of the surface of fouled PA-g-C₃N₄ membrane after cleaning by different methods. (a1) Soak; the first cycle. (a2) Soak; the second cycle. (a3) Soak; the third cycle. (b1) Rinse; the first cycle. (b2) Rinse; the second cycle. (b3) Rinse; the third cycle. (c1) The first cycle of photocatalytic cleaning. (c2) The second cycle of photocatalytic cleaning. (c3) The third cycle of photocatalytic cleaning.

The corresponding results has been added to the Supplementary Figure 21 of the revised supporting information.

Comments 11:

The procedure for membrane preparation does not gives the amounts of materials dissolved in solutions. Are concentrations the same as for the free-standing film? The free-standing film does not mention phosphoric acid, but membrane fabrication does. The reader needs sufficient detail to

reproduce the procedure. What was the effective area of the membranes in the permeation tests? How long did the permeation occur to achieve steady state?

Response:

Thanks a lot for your kind comments. The compositions of the aqueous phase and organic phase were the same for preparing the free-standing films and the composite membranes. Phosphoric acid was added in both cases. The effective area of the membranes in the permeation tests was 26 cm². In order to achieve steady state, the membranes were pre-pressurized for 2 hours under 6 bar.

According to your suggestions, we have added the protocol for membrane preparation including concentrations, detailed procedure, the effective area of the membranes in the permeation tests and stabilization time into the Methods session of the revised manuscript.

The corresponding descriptions are also given below:

“Preparation of freestanding nanofilms. Firstly, 2.0g PIP, 0.5g trimethylamine (TEA) and 0.135 wt% g-C₃N₄ were dissolved in 100 mL water. Secondly, 0.16 mL phosphoric acid was added to adjust the solution pH to 9. Next, a 0.4 wt% TMC n-hexane solution was prepared. Then, the PIP solution was poured into a petri dish. After that, the TMC organic solution was gently added on top of the aqueous solution to induce interfacial polymerization. After 5 min, a freestanding film was formed that was taken out of the petri dish, washed by DI water for three times and stored in DI water.

Preparation of PA and PA-g-C₃N₄ membranes. The PA and PA-g-C₃N₄ NF composite membranes were prepared by IP on top of the polyether sulfone (PES, MWCO 100 kDa) ultrafiltration (UF) membranes. The aqueous solution was prepared with 0.2 to 2.0 g of PIP and 0.5mL TEA in 100 mL water, and its pH was tuned to 9 by adding 0.16 mL phosphoric acid. To prepare the PA-g-C₃N₄ composite membrane, g-C₃N₄ nanosheets were added at the concentrations of 0.075 to 0.145 wt%. The organic solution was prepared by dissolving 0.1 to 0.5 wt% TMC in n-hexane. Then, the PES membrane was dipped into the aqueous phase for 5 min. After being taken out, the excess solution on its surface was removed using a rubbery roller and the support was left at room temperature until the surface appeared dull and dry. Next, the membrane was immersed in the organic phase for 1 min. After that, the membrane was heated in an oven at 80 °C for 5 min, and stored in DI water before use. Three replicate membranes for each experiment were fabricated and examined to get a reproducible performance.’

“Membrane performance test. The permeation performance of the membrane was measured on the cross-flow filtration equipment. The effective area of the membrane was 26 cm². The

solution temperature was maintained at 25°C by a heat exchanger. In order to achieve steady state, the membranes were pre-pressurized for 2 hours under 6 bar. The flow rate was 1.5 L·min⁻¹. The concentrations of salts (including single NaCl, Na₂SO₄) and dyes in feed solutions were 2000 ppm and 100 ppm, respectively.”

Comments 12:

Can the authors give uncertainties in Table S6 and note how many different membranes were used in each case? These and other experiments should employ replicate membranes, not just multiple experiments on the same membranes.

Response:

Many thanks for your kind comments. Three membranes were used in each case. The experiments were conducted with replicate membranes. According to your advice, we have added the uncertainties in original Table S6 (now Table S7) of the revised supporting information.

Comments 13:

If concentration polarization is not large, the membrane rejection will increase with flow rate. Is this a possible reason why the membranes show such high Na₂SO₄ rejection along with high permeability?

Response:

Thank you for the comment. We agree that membrane rejection will increase with flow rate due to reduced concentration polarization. We tested the salt rejection at different flow rates; as shown in Figure R1, the rejection increases only slightly at higher flow rate.

Our tests were done in a round cell (with dimensions given below) at the feed flow rate of 1.5 L min⁻¹. This is high compared to real applications; but is comparable or higher to some extent than literature work with which we compared our performances with in Fig. 4. Since rejection only decreases slightly at lower flow rate, we may conclude that the high selectivity at high permeance is mainly resulted from the unique membrane structure.

Figure R1. Effect of flow rate on Na₂SO₄ rejection.

Figure R2. Illustration of the NF testing cell.

The corresponding results has been added into the Supplementary Figure 20 of the revised supporting information.

Comments 14:

The manuscript suggests that slow spreading “confirms that the diffusion rate of the (PIP+g-C₃N₄)/TMC system is slower than that of the PIP/TMC system.” Spreading and diffusion are not the same. Perhaps the term mixing, rather than diffusion, is appropriate. This seems more of a mixing and spreading phenomenon.

Response:

Many thanks for your suggestion. We agree with you that spreading is not the same as diffusion. However, mixing seems to be improper since PIP reacts with TMC rather than mixing with it. The optical experiment captured the formation of the polyamide layer, which is related to the movement of PIP into the hexane phase. In addition, we do have NMR data to show the changes in the diffusion rate of PIP. We have revised the relevant statements to “which confirms that the mobility of PIP within the (PIP+g-C₃N₄)/TMC system is lower than that with the PIP/TMC system” in the revised manuscript.

Responses to the Reviewer #3:

Present work demonstrates fast permeation and selective ion separation by regulation of interfacial polymerization (IP). g-C₃N₄ is applied for controlled diffusion of PIP by one order of magnitude, thereby forming a nanoscale-ordered hollow cone structure. MD simulation explains the role of g-C₃N₄ in the IP process to understand the role of membrane microstructure in the nanofiltration process. Although I cannot recommend this manuscript for publication in Nature Communications due to lack of novelty, let me advise the following points for the authors' consideration.

Response:

Thank you for the valuable comments and suggestions. We have revised our manuscript accordingly, and a point-to-point reply is provided below. We hope that we have clarified the novelty of our work, and addressed your other concerns.

Comments 1:

Incomplete literature survey: The authors need to survey previously reported papers such as “Synthesis and characterization of g-C₃N₄ nanosheet modified polyamide nanofiltration membranes with good permeation and antifouling properties” and “g-C₃N₄ nanofibers network reinforced polyamide nanofiltration membrane for fast desalination”. I suggest mentioning of these works in the introduction section with emphasizing the novelty of the present work in their respect.

Response:

Thank you for the valuable suggestion. The two papers you mentioned are very helpful. According to your kind advice, we have cited these papers and highlighted the novelty of our work by compared with these papers in our Introduction session.

The novelties of our works mainly include: 1) We provide the way to incorporate the g-C₃N₄ in the IP process to spatially and temporarily modulate the reaction which leads to a nano-scale ordered morphology; 2) The mechanism of the g-C₃N₄ in IP process and the transport mechanism of water molecules are systematically studied using molecular simulation and CFD; 3) Extraordinary separation capacities, in particular, high permeance along with high ion-ion selectivity for value-added separations, have been achieved.

Comparing to reference 1 (“Synthesis and characterization of g-C₃N₄ nanosheet modified polyamide nanofiltration membranes with good permeation and antifouling properties”), there are four major differences: 1) The membranes in ref. 1 after the addition of g-C₃N₄ look rough on the surface, and no nanoscale ordered structure is observed; as a result, the overall membrane

performances after addition of g-C₃N₄ are much lower than our membrane; this might be due to the different way g-C₃N₄ was prepared and different IP protocol that was employed; 2) In ref.1, they did not explain the underlying mechanisms, while MD is employed in our work to provide fundamental understanding of the role of g-C₃N₄ in interfacial polymerization and CFD is used to probe into the transport phenomena; 3) Extraordinary separation capacities, in particular, high permeance along with high ion-ion selectivity for value-added separations, have been achieved in our work; 4) Self cleaning properties were not reported in ref. 1 but in our work. These main differences are the most important features that make our work unique among existing work.

Second, the work in reference 2 (“g-C₃N₄ nanofibers network reinforced polyamide nanofiltration membrane for fast desalination”) is essentially different from ours. g-C₃N₄ was used as the interlayer, neither nanoscale ordered structure nor outstanding selectivity/permeance (comparable to our work) was observed in ref. On the other hand, in our work, g-C₃N₄ was mixed with the PIP solution to tune the diffusion and interface spreading of PIP; as a result, nanoscale ordered structure and outstanding performances are achieved.

Again, we greatly appreciate the comment by the reviewer, and we have revised our manuscript as follows:

‘Notably, g-C₃N₄ has been used in prior studies as nanofillers³² or interlayer³³ during interfacial polymerization. However, this is the first time g-C₃N₄ was used for spatial and temporal modulation of the reaction to get a nano-scale ordered morphology and extraordinary separation capacities, and the first time the role of g-C₃N₄ was systematically studied using molecular simulation. Hence, this research demonstrates a unique approach to overcoming the trade-off between permeability and selectivity, providing insight into the design of ultrapermeable and highly selective NF membranes in water purification, desalination and resource recovery.’

Comments 2:

The section titled “Ultrafast permeation and ion-ion separation” requires more discussion. For example, the authors studied separation performance of PA-g-C₃N₄ as a function of g-C₃N₄ concentration and failed to explain the reason of decreased flux beyond 0.145 % of g-C₃N₄.

Response:

Many thanks for your kind comments. In our experiment we found that beyond 0.145 % of g-C₃N₄, the order of hollow nanoscale surface decreases and correspondingly water transport pathways decreases, which leads to flux decline.

Figure S12. SEM surface morphologies. (a) PA membrane. (b) PA-g-C₃N₄ membrane (0.075 wt%). (c) PA-g-C₃N₄ membrane (0.115 wt%). (d) PA-g-C₃N₄ membrane (0.125 wt%). (e) PA-g-C₃N₄ membrane (0.135 wt%). (f) PA-g-C₃N₄ membrane (0.145 wt%).

The corresponding results has been added into the Supplementary Figure 12 of the revised supporting information. In addition, discussions were added in Supplementary session 4.1 to provide explanations on the phenomena:

“Supplementary Table 7 gives the permeance comparison of the PA-g-C₃N₄ membranes with different g-C₃N₄ concentrations. With the increase of g-C₃N₄ concentration, the permeance gradually increases, reaching the maximum at 0.135 wt% g-C₃N₄. At 0.145 wt% g-C₃N₄, the permeance decreases again, which is consistent with the less regular nanoscale structure captured in Supplementary Figure 12.”

Comments 3:

In page 9, line 3, the authors mention that the Cl⁻/SO₄²⁻ selectivity reaches 130 without a technical detail. Experiments carried out for the selectivity measurement need be provided.

Response:

Thank you for the comment. In our original submission, we measured the rejections to NaCl and Na₂SO₄ individually and calculate the selectivity based on Eq. (3). In the revised manuscript, we have done additional experiment by measuring the ion rejections with mixed salt solutions. Specifically, mixed salt tests were performed using the mixed solutions of NaCl and Na₂SO₄, where the total concentrations were kept constant at 2000 ppm while the ratios of were varied. The Cl⁻ and SO₄²⁻ concentrations were measured by ion chromatography. The Cl⁻/SO₄²⁻ selectivity was calculated based on Eq. (3). Details of the experiment were given in the Methods session of the revised manuscript.

The results from mixed salt test were provided in Supplementary Figure 18. The Cl⁻ rejection is around 20%, and the rejection to SO₄²⁻ is > 99% in all tests. The Cl⁻/SO₄²⁻ selectivity can reach up to 130. The high selectivity is attributed to the negative surface charge on membranes and the suitable pore size. Fig. 4f compares our membranes with other membranes reported in literature. The highlight of our work is boosting the membrane permeance while maintaining high selectivity.

Figure S18. Separation performance of the weight ratio of Cl⁻/SO₄²⁻ in 2000 ppm mixed solution. (a) Rejection. (b) Cl⁻/SO₄²⁻ selectivity.

We explained in the manuscript as follows: “Fig. 4d shows that the MWCO of PA-g-C₃N₄ membrane is 472 Da, which is slightly larger than that of PA membrane and is consistent with earlier discussions. A narrow pore size distribution with pore radius of 0.364 nm is achieved, which lies right between the sizes of hydrated Cl⁻ ions (0.332 nm) and SO₄²⁻ ions (0.379 nm) (Fig. 4e). It is also revealed earlier that the membrane surfaces are highly negatively charged. The membrane is thus able to achieve high selectivity of 130 for Cl⁻ over SO₄²⁻ (Fig. 4f) through both size sieving and Donnan exclusion effects.”

Comments 4:

The authors presented the self-cleaning property of PA-g-C₃N₄ as one of the highlights of the work whereas they do not demonstrate or explain the mechanism.

Response:

Thanks a lot for your comment. The reaction between g-C₃N₄ and organic species has been studied in literature (refs, 37, 38). According to your kind advice, we have demonstrated the mechanism figure as the following. As shown in the figure below, the related mechanism is that electrons (e⁻) and holes (h⁺) are generated by visible light with g-C₃N₄. The generated e⁻ can react with dissolved oxygen (O₂) to form superoxide radical anion O₂^{-•}. Then h⁺ and O₂^{-•} degrade the MB to CO₂ and H₂O, membrane surface becomes clean again.

Figure S22. The mechanism of photocatalytic degradation of methylene blue on the PA-g-C₃N₄ membrane.

The corresponding results has been added into the Supplementary Figure 22 of the revised supporting information, the corresponding descriptions have been added into the revised manuscript as the following.

“Electrons (e⁻) and holes (h⁺) are generated by visible light that partially penetrate through the polyamide layer to g-C₃N₄³⁷, which then react with dissolved oxygen (O₂) to form superoxide radical anion O₂^{-•}³⁸. MB is degraded by h⁺ and O₂^{-•} to form CO₂ and H₂O, and thus membrane surface becomes clean again.”

[37] Zhang, L, L. et al. High flux photocatalytic self-cleaning nanosheet C₃N₄ membrane supported by cellulose

nanofibers for dye wastewater purification. *Nano Research*, 14(8), 2568-2573 (2021).

[38] Lei, Z. D. et al. The influence of carbon nitride nanosheets doping on the crystalline formation of MIL-88B(Fe) and the photocatalytic activities. *Small*. 14, 1802045 (2018).

Comments 5:

Self-cleaning properties of PA and PA-g-C₃N₄ need be compared and emphasized. At the same time, the authors may want to consider visual demonstration of the anti-fouling behavior of PA and PA-g-C₃N₄ via SEM images of PA and PA-g-C₃N₄ before and after the fouling tests.

Response:

Many thanks for your kind comment. According to your kind advice, self-cleaning properties of PA and PA-g-C₃N₄ has been compared and shown in Supplementary Figure 20. The results show that self-cleaning properties of PA-g-C₃N₄ membrane is better than that of PA membrane. we have done the anti-fouling behaviour of PA and PA-g-C₃N₄ via SEM images of PA and PA-g-C₃N₄ before and after the fouling tests with light irradiation for each round. It can be seen that the anti-fouling behaviour of PA-g-C₃N₄ membrane was better than that of PA membrane. The corresponding results has been added into the revised supporting information.

Figure S20. SEM images of PA and PA-g-C₃N₄ before and after the fouling tests. (a1) Uncontaminated PA membrane. (a2) Contaminated PA membrane. (a3) First-round photocatalytic PA membrane. (a4) Second-round photocatalytic PA membrane. (a5) Third-round photocatalytic PA membrane. (b1) Uncontaminated g-C₃N₄ membrane. (b2) Contaminated g-C₃N₄ membrane. (b3) First-round photocatalytic g-C₃N₄ membrane. (b4) Second-round photocatalytic g-C₃N₄ membrane. (b5) Third-round photocatalytic g-C₃N₄ membrane.

REVIEWER COMMENTS

Reviewer #1 (Remarks to the Author):

The authors have addressed most of my questions for my comments on first version.

Reviewer #2 (Remarks to the Author):

The authors have thoroughly addressed all my concerns in the revision. I recommend publication.

Reviewer #3 (Remarks to the Author):

The authors addressed my concerns with some elaborate experiments, though my concern about the conceptual novelty of the work suitable for publication in Nature Communications remains unaltered. Let me list below point by point the reviewer's comments and the author's responses.

Comments 1:

(Incomplete literature survey) The authors need to survey previously reported papers such as "Synthesis and characterization of g-C₃N₄ nanosheet modified polyamide nanofiltration membranes with good permeation and antifouling properties" and "g-C₃N₄ nanofibers network reinforced polyamide nanofiltration membrane for fast desalination".

I suggested mentioning these works in the introduction section emphasizing the novelty of the present work in their respect.

The authors did an elaborate comparison of the present work with the above-mentioned literature. They highlighted the superiority of their work based on performance.

However, my concern remains unaltered with the novelty of the present work. Despite the advanced performance, the present work fails to establish novelty in terms of the conceptual uniqueness of the subject matter or the application.

Comments 2:

The section titled "Ultrafast permeation and ion-ion separation" requires more discussion. For example, the authors studied the separation performance of PA-g-C₃N₄ as a function of g-C₃N₄ concentration and failed to explain the reason for decreased flux beyond 0.145 % of g-C₃N₄.

The authors have addressed the reason behind decreased flux beyond 0.145 % of g-C₃N₄ with a supporting experiment.

Comments 3:

On page 9, line 3, the authors mention that the Cl^{sup-}/SO_4^{sup-2} selectivity reaches 130 without a technical detail. Experiments carried out for the selectivity measurement need to be provided.

The authors provided an experimental result to show the Cl^{sup-}/SO_4^{sup-2} selectivity which is now easier for readers to interpret.

Comments 4:

The authors presented the self-cleaning property of PA-g-C₃N₄ as one of the highlights of the work whereas they do not demonstrate or explain the mechanism.

The authors demonstrated the mechanism of self-cleaning with an elaborate schematic for readers. I would like to mention here that the self-cleaning property of C₃N₄ membranes is well established in the literature, which leads to questioning the novelty of the work for publication in Nature Communications.

Comments 5:

The self-cleaning properties of PA and PA-g-C₃N₄ need to be compared and emphasized. At the same time, the authors may want to consider a visual demonstration of the anti-fouling behavior of PA and PA-g-C₃N₄ via SEM images of PA and PA-g-C₃N₄ before and after the fouling tests.

The authors demonstrated a better anti-fouling behavior of PA-g-C₃N₄ as compared to that of PA membranes via SEM images for 3 rounds of photocatalytic reaction.

Reply to Comments of Reviewers

Responses to the Reviewer #1:

The authors have addressed most of my questions for my comments on first version.

Response:

Many thanks for your positive evaluation of our work.

Responses to the Reviewer #2:

The authors have thoroughly addressed all my concerns in the revision. I recommend publication.

Response:

Many thanks for your positive evaluation of our work.

Responses to the Reviewer #3:

The authors addressed my concerns with some elaborate experiments, though my concern about the conceptual novelty of the work suitable for publication in Nature Communications remains unaltered. Let me list below point by point the reviewer's comments and the author's responses.

Response:

Thank you for your valuable suggestions which have helped a lot in improving our manuscript. We believe our work has presented sufficient novelty in the way that this is the first conceptual demonstration of simultaneous temporal and spatial control of the interfacial polymerization process to obtain nanoscale ordered structure. High permeance and selectivity are bonus points. We have provided more discussions in the point-to-point reply in the next.

Comments 1:

Incomplete literature survey: The authors need to survey previously reported papers such as

“Synthesis and characterization of g-C₃N₄ nanosheet modified polyamide nanofiltration membranes with good permeation and antifouling properties” and “g-C₃N₄ nanofibers network reinforced polyamide nanofiltration membrane for fast desalination”. I suggest mentioning of these works in the introduction section with emphasizing the novelty of the present work in their respect.

The authors did an elaborate comparison of the present work with the above-mentioned literature. They highlighted the superiority of their work based on performance.

However, my concern remains unaltered with the novelty of the present work. Despite the advanced performance, the present work fails to establish novelty in terms of the conceptual uniqueness of the subject matter or the application.

Response:

Thank you for your comments. The two papers you mentioned are very helpful. In our last version, we highlighted the superiority of our work as follows ‘Notably, g-C₃N₄ has been used in prior studies as nanofillers³² or interlayer³³ during interfacial polymerization. However, this is the first time g-C₃N₄ was used for spatial and temporal modulation of the reaction to get a nano-scale ordered morphology and extraordinary separation capacities, and the first time the role of g-C₃N₄ was systematically studied using molecular simulation.’ Instead of emphasizing the performance, we highlighted that the major novelty of the work is the first conceptual demonstration of simultaneous spatial and temporal control of interfacial polymerization to get a nanoscale ordered structure. This is also the first time the role of g-C₃N₄ was systematically studied using molecular simulation. In addition, in terms of performance which is a bonus point of this work, we focused on the permeance and solute-solute selectivity tradeoff. Although this is not a completely new application, solute-solute selectivity is indeed less touched by similar studies in the field.

We looked further into the two references. The differences between our work and the two references have been clearly elaborated in the last point-to-point reply. Ref. 33 used g-C₃N₄ as the interlayer, which is completely different from ours. Ref. 32 also used g-C₃N₄ as the nanofillers. In addition to the differences we mentioned last time, we tried to dig more into the underlying reasons. We noticed that there are perhaps a few more critical differences. First, the concentration of g-C₃N₄ in our work is 50 times higher than that in ref. 32; this high nanosheet concentration likely accounts for the spatial and temporal effects of g-C₃N₄ that lead to ordered hollow structure. In Supplementary Figure 12, it is shown that the nanoscale hollow structure becomes more ordered and significant at high g-C₃N₄ concentrations. This may be the most important difference. Second, the methods to perform interfacial polymerization are different; the control membrane in our work shows much better performances (and we have justified the validity of the results in the reply to

editor). Third, the pH of the solution was adjusted, which may affect the interfacial polymerization process as well as the properties of g-C₃N₄. We therefore made some revisions to our introduction: ‘Notably, g-C₃N₄ has been used in prior studies as nanofillers³² or interlayer³³ during IP. In this work, we prepared g-C₃N₄ suspensions with high concentrations that induce spatial and temporal modulation to get a nanoscale ordered morphology and extraordinary separation capacities, and the role of g-C₃N₄ was systematically studied using molecular simulation. Hence, this research demonstrates an approach to realizing simultaneous temporal and spatial control of reactions and to overcoming the trade-off between permeability and solute-solute selectivity. It provides insight into the design of ultrapermeable and highly selective NF membranes in water purification, desalination and resource recovery.’

Comments 2:

The section titled “Ultrafast permeation and ion-ion separation” requires more discussion. For example, the authors studied the separation performance of PA-g-C₃N₄ as a function of g-C₃N₄ concentration and failed to explain the reason for decreased flux beyond 0.145 % of g-C₃N₄.

The authors have addressed the reason behind decreased flux beyond 0.145 % of g-C₃N₄ with a supporting experiment.

Response:

Thank you for your positive comment.

Comments 3:

On page 9, line 3, the authors mention that the Cl⁻/SO₄²⁻ selectivity reaches 130 without a technical detail. Experiments carried out for the selectivity measurement need to be provided.

The authors provided an experimental result to show the Cl⁻/SO₄²⁻ selectivity which is now easier for readers to interpret.

Response:

Thank you for your positive comment.

Comments 4:

The authors presented the self-cleaning property of PA-g-C₃N₄ as one of the highlights of the work whereas they do not demonstrate or explain the mechanism.

The authors demonstrated the mechanism of self-cleaning with an elaborate schematic for readers. I would like to mention here that the self-cleaning property of g-C₃N₄ membranes is well

established in the literature, which leads to questioning the novelty of the work for publication in Nature Communications.

Response:

Thanks a lot for your comment. We agree that the self-cleaning property of g-C₃N₄ nanosheet is established. The discussions are still included in this manuscript because: 1) fouling is a critical challenge in water treatment processes and strategies to mitigate the issue are important; and 2) though different types of self-cleaning g-C₃N₄ membrane were reported in literature, this is the first demonstration of g-C₃N₄ -PA membrane with self-cleaning capability. Nonetheless, we do agree that the self-cleaning is not sufficiently novel and it was not mentioned as a highlight in the introduction. Instead, the conceptual demonstration of simultaneous spatial and temporal control of reaction is the major innovation in this paper. We have also now deleted the relevant statement from the abstract:

~~The membrane also has self-cleaning capabilities.~~

Comments 5:

The self-cleaning properties of PA and PA-g-C₃N₄ need to be compared and emphasized. At the same time, the authors may want to consider a visual demonstration of the anti-fouling behavior of PA and PA-g-C₃N₄ via SEM images of PA and PA-g-C₃N₄ before and after the fouling tests.

The authors demonstrated a better anti-fouling behavior of PA-g-C₃N₄ as compared to that of PA membranes via SEM images for 3 rounds of photocatalytic reaction.

Response:

Thank you for your positive comment.